# Health Promotion Actions and School Violence—A Cluster Analysis from Finnish Comprehensive Schools

**DOI:** 10.3390/ijerph191912698

**Published:** 2022-10-04

**Authors:** Noora Ellonen, Miko Pasanen, Kirsi Wiss, Laura Mielityinen, Elina Lähteenmäki, Katja Joronen

**Affiliations:** 1Faculty of Social Science, Tampere University, 33014 Tampere, Finland; 2Department of Nursing Science, University of Turku, 20014 Turku, Finland; 3Finnish Institute for Health and Welfare, 00271 Helsinki, Finland

**Keywords:** school violence, health promotion, cluster analysis, school characteristics

## Abstract

(1) Schools have a significant role in violence prevention activities. This study aimed to first identify profiles of Finnish comprehensive schools based on school violence. The second aim was to examine the associations between profiles concerning health promotion actions, reactive or punitive actions, and school characteristics. (2) The study used the large-scale, nationally representative Benchmarking System of Health Promotion Capacity-Building (BSHPCB) data (*n* = 2057 schools) completed by the school’s principal together with a student welfare team. The data was analyzed by cluster analysis and Chi-squared and Kruskal–Wallis tests. For post hoc testing, Fisher’s exact test with odds ratios and Mann–Whitney U-test were used. (3) The cluster analysis yielded five profiles of school violence: “No violence”, “Adolescent violence” (violence both among pupils and from pupils towards staff, but not inappropriate behavior from school staff towards pupils), “Not known” (principals either did not respond to these questions or they did not know whether there had been any school violence incidents), “Peer violence” (school violence occurred among pupils but not from pupils towards staff, nor inappropriate behavior from school staff towards pupils), and “All violence” (all types of school violence and inappropriate behavior from school staff towards pupils). These clusters differed according to type of school and municipality. Additionally, both management and monitoring as health promotion actions were related to higher incidence of school violence whereas other actions, such as commitment, resources, common practices, and participation were not related to school violence. (4) The findings of this study indicate that schools have different profiles in terms of school violence and providing evidence and guidance for school violence prevention work.

## 1. Introduction

School violence is defined as youth violence that occurs on school property, on the way to or from school or school-sponsored events, or during a school-sponsored event. A young person can be a victim, perpetrator, or witness of school violence [1]. Schools have a significant role in violence prevention activities. First, the provision of education and organized activities helps protect against violence. Second, schools can challenge harmful social and cultural norms that tolerate violence and key risk factors for violence, such as substance use. Third, school plays an important role in preventing violence by developing children’s and young people’s life skills as well as teaching safe behavior and how to protect themselves from abuse [2]. Fourth, modifying school environmental factors can reduce violent behavior. School violence prevention programs should therefore focus on both the individual and the environment. One should also think about the role of the school in situations of violence, rather than focusing on the perpetrator of violence. The school plays a significant role in educational and social guidance for children and young people [3]. Increasing research evidence suggests that school climate plays a meaningful role in school violence as positive school climate lowers risk behavior [4] and is associated with lower rates of school victimization [5].

School safety is defined by United Nations Educational, Scientific and Cultural Organization [6] as the process of establishing, and maintaining, a school that is a physically, cognitively, and emotionally safe space for students and staff to carry out learning activities. Cohen (2021) summarizes research-based policy and practice steps that can increase school safety. Those steps increasing school safety are, for example, (1) improving positive relationships between students and teachers, (2) helping the student population to believe in school rules and their fairness, (3) supporting coordinated educational interventions, (4) ensuring a classroom and school environment that is positive and focuses on understanding students, and (5) implementing school safety interventions that are focused on improving the physical environment of the school, especially reducing the amount of perceived school physical disorder [7].

Studies have shown that prevention efforts can reduce violence at school and improve the school environment. Teachers, administrators, students, family members, and friends can participate in these efforts [8] and play a key role in making students feel safe at school [9]. Bystanders, as well as teachers, have the potential to stop violence once it has started and provide support to victims in the aftermath of a violent incident. They may also prevent school violence (i.e., stalking and sexual harassment) from occurring in high risk situations by shifting social norms and promoting positive bystander action [10]. While the research evidence suggests that some bystander-interventions are helpful, not all bystander action is experienced as beneficial by the survivors of school violence or may not make a difference. Sometimes it can even be harmful [11].

A body of research exists on promoting individual and organizational health to prevent school violence or to intervene when it occurs [7]. There is contradictory research evidence about whether school size, class size, school level, school location, percentage of male students, type of school, and school poverty have an impact on school violence. Crooks et al. (2007) found no significant relationship between school size and student violence behavior [12]. Similarly, Khoury-Kaassabri et al. reported no significant association between school size and physical victimization, threats, verbal-social victimization, or property damage [13,14]. In Agnich et al.’s (2013) study, school size was not included in the final model but mean school size was positively related to the level of violence at the national level and class size was positively related to principals’ reports of school violence [15]. The contradictory findings of Brookmeyer et al.’s (2006) study revealed that a larger school significantly predicted future violent behavior [16]. Wilcox et al. (2006) indicated that a larger school was associated with more teacher victimization [17]. Several previous study findings [12,16,18] indicated no association between school location and school violence. For instance, a Canadian study about the link between childhood maltreatment and violent delinquency showed that school location (rural vs. urban) had no significant impact on student violence behavior [16]. However, in the US, urban teachers reported a greater probability of witnessing school violence [11], and Crawford and Burns (2015) noted that school location offered more explanatory power with regard to various types of school violence [19]. Astor et al. (1999) suggested that one reason for that is schools’ structure. In the US, urban schools are built vertically, and there are a lot of unsupervised areas where students find ways to misbehave [20].

School nurses recognize their role in school violence, but no evidence of school nurses’ role in interventions to reduce serious youth violence has been found [21]. Stephens and Sayer (2021) suggest that staff members experience low confidence in delivering universal education on youth violence, and they want and need further support and guidance [21].

Supportive and punitive school practices are usually seen as opposite ways to respond to school violence. According to Crawford and Burns (2020), the supportive school responses revealed some promising findings in reducing school violence, whereas punitive responses had no effect on or even increased school violence [22]. A review article by Mallett (2016) supports this finding [23]. Johnson’s (2009) review indicated that school norms against violence were associated with a decrease in student-reported perpetration and victimization. The same review also showed that school norms about violence have been studied to a greater extent than other school social environment measures [24]. There is a lack of research on associations between school-based health promotion actions and school violence; thus, this study contributes to school research from the health-promotive perspective on school violence.

The aim of this study was twofold. First, we identified profiles or subgroups of comprehensive schools based on school violence using the nationally representative Benchmarking System of Health Promotion Capacity-Building (BSHPCB) data. Second, we examined the associations between profiles concerning *health promotion actions* and *school characteristics*. Health promotion actions were defined through seven dimensions, i.e., commitment, management, monitoring and needs assessment, resources, common practices, participation, and other core functions. School characteristics referred to school type, type of municipality, and student count.

### Context of the Study

The educational system in Finland is unique as, unlike in many other countries, basic education (i.e., primary, lower secondary and comprehensive school) is free and compulsory for everyone. Schools are also mostly public and maintained by municipalities [25]. The teaching is based on a thorough nationwide curriculum in all schools [26], so the quality of education should be equal in all schools. This provides a uniform education for the whole age group, and consequently pupils usually go to the nearest institution without any selection procedure. Despite all the homogeneity, there are still some differences between schools, which are mostly explained by the school characteristics, such as the diversity of pupil body and the resources offered by the municipalities [25,27].

Finnish Pupil Welfare is also a rarity because it is statutory; it is a free service in all schools. Therefore, schools in Finland play a significant role in health promotion. Communal pupil welfare is primarily preventive, and it includes collaborative monitoring and needs assessments, such as School Health Surveys, extensive health inspections, three-year inspections of community well-being and environmental safety, and the implementation of the Student Welfare plan to protect pupils from violence. The Student Welfare plan covers both preventive and reactive actions [26,28]. Preventive actions are always primary; reactive, mainly disciplinary, means, such as disciplinary educational discussion and temporary expulsion, are used only if needed [29,30]. In addition, as a management action, a multidisciplinary Student Care Team operates actively in every school. Its main task is to plan, implement, and evaluate the work of communal pupil welfare services in cooperation with pupils and their households [31]. Pupils are also entitled to individualized pupil welfare, which consists of services provided by school health nurses, school doctors, curators and psychologists, and these resources are allocated to pupils and their guardians directly [26].

## 2. Materials and Methods

The data was gathered from the BSHPCB data collection in comprehensive schools with grades 1 to 9 in 2019. The BSHPCB is a nationwide benchmarking tool for local governments and schools to manage, plan, and evaluate their own health promotion activities and resources in basic education. The BSHPCB is run by the Finnish Institute for Health and Welfare (THL), and the data collection in basic education is done in collaboration with the Finnish National Agency for Education. The data was collected between October and December 2019, and 91% (*n* = 2057) of comprehensive schools in Mainland Finland participated.

Procedure:

The data collection was sent in October 2019 to all comprehensive schools with grades 1 to 9 in mainland Finland. It was addressed to principals of schools (N = 2268). The electric data collection form was requested to be compiled in cooperation between the school’s principal and the student welfare team. Data were received by the end of December 2019 from 2057 (91%) schools.

### 2.1. Measures

School violence was measured using three questions in BSHPCB data collection form. The questions were: During the school year 2018–2019, did the following situations occur in your school: (1) Pupil threatened school personnel with violence or was violent towards them, (2) Violence between pupils, and (3) Personnel’s inappropriate behavior towards a pupil (e.g., violence, harassment, bullying). The response options were (1) No; (2) Yes, not documented; (3) Yes, documented; (4) Yes, documented and number data was summarized; and (5) Unknown. School characteristics were measured by three questions concerning *School type, Type of municipality, and Student count* (see Table 1).

Health promotion actions in schools were measured by the BSHPCB data collection form including seven themes: (1) Commitment, (2) Management, 3) Monitoring and needs assessment, (4) Resources, (5) Common practices, (6) Participation, and (7) Other core functions. In this study we used the following questions in terms of the themes:(1)**Commitment***(Student participation in school meals)*;(2)**Management** *(Pupil welfare group meetings; Monitoring of absences; Check on the health and safety of school environment and welfare promotion among the learning community)*;(3)**Monitoring and needs assessment** *(Monitoring of bullying, Monitoring of smoking and alcohol and drug use, Monitoring of disciplinary procedures, Monitoring of accidental injuries, Health and welfare monitoring, Reporting of data on pupil health and well-being, Content of the comprehensive health assessment, Monitoring of absences)*;(4)**Resources***(Teacher resources, Classroom assistant resources, School health nurse resources, School physician resources, School psychologist resources, School social work resources)*;(5)**Common practices** *(Prevention of smoking and alcohol and drug use, Prevention of harassment and violence, Processing of bullying incidents, Absences, Prevention of accidents and injuries, Guidelines, School meals)*;(6)**Participation** *(Parent association and peer pupil scheme, Parents’ possibilities to influence, Pupils’ possibilities to influence, Description of pupil welfare services on school website/in school brochure, Home-school collaboration)*;(7)**Other core functions** *(Measures to promote physical activity during the school day, Participation in school meals)*.

Table 1 presents the school characteristics and school violence in comprehensive schools. Most of the participating schools were primary schools grades 1–6 (61%, *n* = 1249) and urban schools (53%, *n* = 1204). The median of the student count in all participating schools was 174 pupils but the range was large (3–1374).

### 2.2. Data Analysis

Data were analyzed statistically using R version 4.0.2. First, a cluster analysis was conducted to find out whether schools differed in terms of school violence. The cluster analysis of the schools was performed using questions about three scenarios: “Pupil threatened school personnel with violence or was violent towards them”, “Violence between pupils” and “Personnel’s inappropriate behavior towards a pupil (e.g., violence, harassment, bullying”). Each of these questions had three possible categories: yes, no, or unknown (no response or not known). Distance between the datapoints was measured using Gower’s distance [32], and clustering was performed using partitioning around medoids. The average silhouette width of the clusters was used to aid selection of the appropriate number of clusters [33].

After forming clusters, statistical testing using Chi-squared and Kruskal–Wallis tests was performed to test the differences between the clusters in all other variables included in the analysis. Due to the large sample size, effect size was used to choose the variables for which post hoc testing was performed (Cohen’s w > 0.2 or eta squared > 0.06). For post hoc testing, Fisher’s exact test with odds ratios and Mann–Whitney *U*-test were used. The results of this were presented in a heatmap, which contains odds ratios for all the pairwise comparisons (see Figure 1 Heatmap).

## 3. Results

### Clusters’ Description

Over 70 percent (71%) of the schools reported violence between pupils. Moreover, over half of the schools (54%) reported that pupils threatened school personnel with violence or were violent towards them. Fourteen percent of the participating schools declared personnel’s inappropriate behavior towards a pupil, such as violence, harassment, or bullying. A considerable number of schools (ranging from 12% to 16%) did not report or did not know whether or not school violence occurred in the school. Table 2 presents the reported prevalence of school violence in participating schools.

The cluster analysis identified five groups of schools (Table 3). The *first* cluster was composed of schools in which there was very little school violence of any type or inappropriate behavior from school staff towards pupils. The cluster is named *“No Violence”*. The first cluster included a total of 383 schools (19% of all schools) and the median number of pupils was 63 (range: 3–997), indicating small schools. The *second* cluster was composed of schools in which school violence, both among pupils and from pupils towards staff, was reported in all schools. However, inappropriate behavior from school staff towards pupils was not reported in any of these schools. This cluster was named *“Adolescent violence”*. This cluster was the largest, including 851 schools (41% of all schools), and the median number of pupils was 247.5 (range: 6–1134), indicating moderately large schools.

The *third* cluster, *“Not known”*, was composed of schools in which incidents of school violence were unknown, meaning that principals either did not respond to these questions or they did not know whether there had been any school violence incidents. This cluster was excluded from further analysis. The *fourth* cluster was composed of schools in which school violence occurred among pupils but not from pupils towards staff. Neither there was inappropriate behavior from school staff towards pupils. This cluster was named *“Peer violence”*. This cluster included 445 schools (22% of all schools) and the median number of pupils was 126 (range: 15–738), indicating small schools. Finally, the *fifth* cluster was composed of schools in which there were all types of school violence and inappropriate behavior from school staff towards pupils. This cluster was named *“All violence”*. This cluster included 332 schools and the median number of students was 351 (range 21–1374), indicating large schools.

Most schools in clusters 1, 2, and 4 were comprehensive primary schools (grades 1–6, ages 7–13), whereas most schools in cluster 5 were comprehensive schools (grades 1–9, ages 7–16). Most schools in clusters 2, 4, and 5 were urban area schools, whereas in cluster 1, where there was very little or no school violence, schools were mostly in rural or suburban areas. (Figure 2).

The heatmap in Table 2 indicates odds ratios for the pairwise comparisons, including variables with high enough effects size in our preliminary analysis (Cohen’s w > 0.2 or eta squared > 0.06). The columns of the heatmap represent cluster pairs, each row is one pairwise comparison of different variables. For example, the first row, first column estimates whether there are higher odds that the school in question is in cluster 2 than cluster 1 if the school’s type of municipality is rural compared to urban. If OR > 1 then the odds are higher for cluster 2, given the rural municipality compared to rural municipality. If OR < 1 odds are higher for cluster 1, given the rural municipality compared to rural municipality. The pairwise comparisons revealed that both management and monitoring as health promotion actions were related to higher incidence of school violence, whereas other actions, such as commitment, resources, common practices, and participation did not.

In terms of management as health promotion action, odds ratios for monthly pupil welfare group meetings were higher in schools with school violence incidents compared to schools with no school violence (cluster 1). In addition, odds for more regular meetings were higher in schools with violence from pupils towards other pupils and towards staff compared to schools with school violence only among pupils. The same finding can be seen in organizing educational discussion.

In terms of monitoring health promotion action, the use of tobacco and suspected use of drugs seem to follow the occurrence of violence. In schools with all types of school violence (cluster 5, All violence), odds ratios for the use of tobacco and suspected use of drugs were higher. This was seen also in use of tobacco by school staff.

Odds for different kinds of punishments for pupils were clearly higher in schools with school violence. The difference is clearest between schools in cluster 5, with all types of school violence, and schools in cluster 1 with no violence. However, odds for punishments were also higher in schools in clusters 2 (Adolescent violence) and 4 (Peer violence) compared to schools in cluster 1 (No violence). Further, odds for all types of punishments were higher in schools with pupils’ violence towards other pupils and school personnel (cluster 2), compared to schools with violence only among pupils (cluster 4).

## 4. Discussion

The primary aim of this study was to identify profiles of Finnish comprehensive schools based on school violence. Our analysis revealed that different school profiles in terms of school violence can be found among Finnish schools, while five distinct clusters of schools were identified: (1) No violence, (2) Adolescent violence, (3) Not known, (4) Peer violence, and (5) All violence. Adolescent violence refers here to violence from pupils towards other pupils or towards teachers, whereas peer violence refers to violence between adolescents. These differences are interesting in the context of Finnish schools, which used to be seen as very homogeneous, although in recent research other differences have also been reported [27].

The third cluster (Not known), where 257 schools reported “not known” to all violence questions were excluded from further analysis, but this is an interesting cluster from the discussion point of view. “Not known” refers here to situations in which respondents did not want to respond to any violence-related questions or they did not know if there had been any violence in their school. From the school health promotion point of view, both responses raise concerns. First, school surveys are very important tools in developing health promotion nationally, and school personnel should feel obligated to provide information to inform that work. Second, not knowing whether or not there is violence at one’s school reflects serious problems in communication between pupils and personnel.

Overall, school violence is very common in Finnish schools, and violence between pupils seems to be the most common type. However, over half of the schools also reported that pupils threatened school personnel with violence or were violent towards them. School violence is more common in comprehensive schools (grades 1–9, ages 7–16) and in urban area schools compared to primary comprehensive schools (grades 1–6, ages 7–13) and schools in rural or suburban areas.

The secondary aim was to examine the associations between profiles concerning health promotion actions, reactive/punitive actions, and school characteristics. Our analysis suggests, first, that there are several actions or practices in schools that are not related to violence in schools, such as human resources or actions for students’ or parents’ participation.

Second, pupil welfare meetings as well as educational discussions seem to be more reactive to violence than preventive. Overall, in school health promotion activities, preventive actions should be primary [29,30], but this analysis also suggests that actions originally planned to be preventive are mainly used reactively. Although all these actions are regulated by laws and curriculum, implementing them is the responsibility of the schools and the municipalities they are in, which may cause policies to be implemented in a different way than was intended. This is an important issue for future research because the equality of services and prevention actions pupils receive is an important aspect of the Finnish school system.

Third, disciplinary educational discussions seem more common in schools with violence towards personnel compared to schools with only peer violence. This may indicate that violence towards personnel is taken more seriously than violence among pupils. This finding should be studied further.

Preventing school violence requires addressing all the factors that put people at risk or protect them from violence. World Health Organisation in collaboration with UNESCO and UNICEF highlight the nine activities to school-based violence prevention: (1) Develop leadership, school policies, and coordination methods; (2) Collect data on violence and monitor changes over time; (3) Prevent violence through curriculum-based activities; (4) Work with teachers on values and beliefs and train them in positive discipline and classroom management; (5) Respond to violence when it happens; (6) Review and adapt school buildings and grounds; (7) Involve parents in violence prevention activities; (8) Involve the community in violence prevention activities; and (9) Evaluate violence prevention activities and use the evidence to strengthen your approach [34].

## 5. Strengths and Limitations 

This study has several strengths. First, the data is nationally representative and it covers 91% of schools in mainland Finland. Second, the BSHPCB data collection form has been used since 2009 that supports the reliability of the questionnaire. The strengths and development areas of the data collection are reviewed after each data collection. Before each new data collection, the form is reviewed and developed together with national actors and schools’ principals. Each data collection process also includes data auditing. Some limitations of this study should also be mentioned. First, it is possible that the respondents do know recall exactly the all the dimensions of the BSHPCB. Second, the intensity of health promotion activities could have been assessed in more detail.

## 6. Conclusions

This study contributes to research on school violence through school-based actions. There are few nationally representative school-level studies that explore school violence from a comprehensive point of view and that seek explanations from the large variety of school characteristics and preventive actions.

The results of our study indicated that school violence, especially violence between pupils, is very common in Finnish schools. Additionally, over half of the schools also reported that pupils threatened school personnel with violence or were violent towards them. We also found that school violence is more common in comprehensive schools (grades 1–9, ages 7–16) and in urban area schools compared to primary comprehensive schools (grades 1–6, ages 7–13) and schools in rural or suburban areas. Thus, schools need to focus on structural, functional, and socio-emotional dimensions when developing violence prevention and safety promotion in school communities. Our study revealed that there are several school-based actions or practices that are not related to violence in schools, such as human resources or actions for students’ or parents’ participation. Further studies are needed to explore in more detail the frequency and intensity of school-based actions on violence prevention. Results may be used by politicians, teachers, and other school stakeholders as a starting point to rethink school policies and to develop more promotive actions. School stakeholders should be able to obtain and use the information about the school violence profiles regularly. Future studies are needed to analyze in more detail the effectiveness of health promotion actions on school safety and violence.

## Figures and Tables

**Figure 1 ijerph-19-12698-f001:**
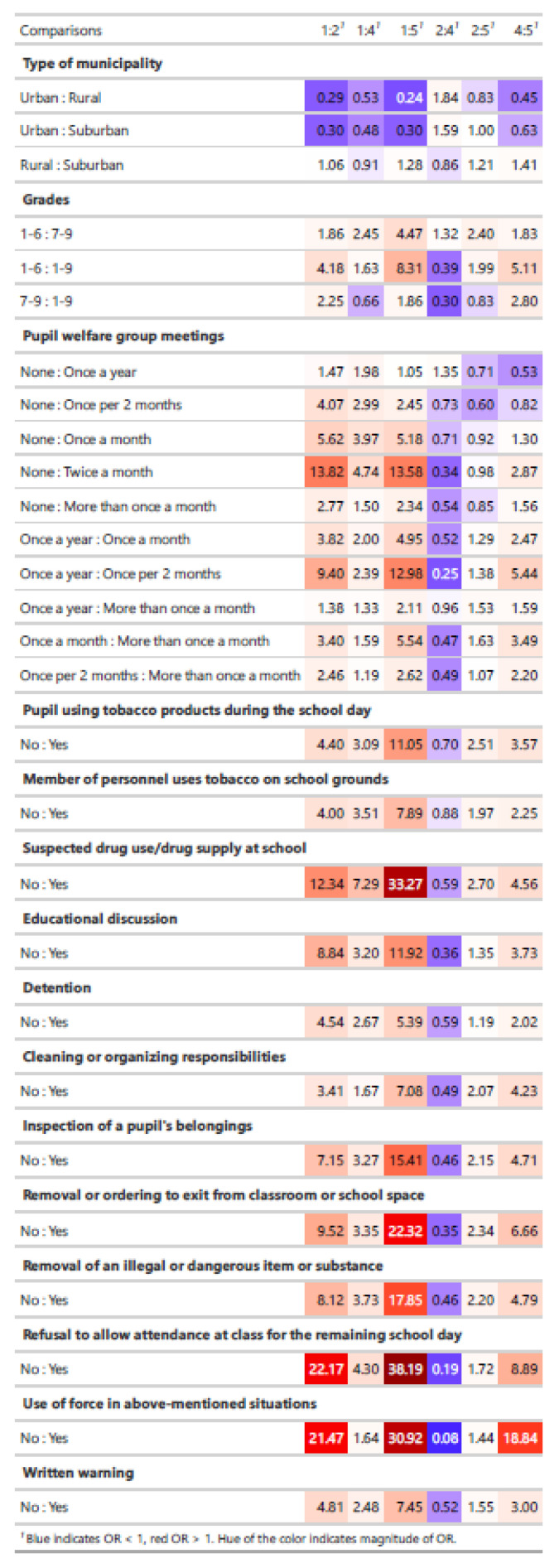
Heatmap indicating odds ratios for the pairwise comparisons.

**Figure 2 ijerph-19-12698-f002:**
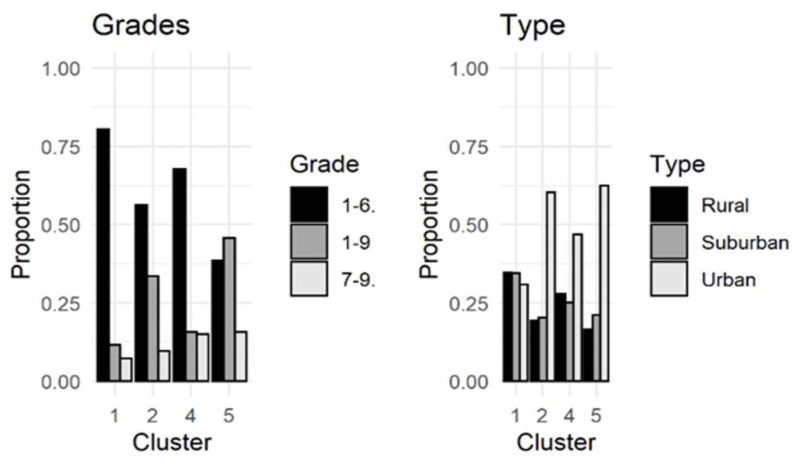
Proportion of type of school and municipality within clusters (*n* = 2268).

**Table 1 ijerph-19-12698-t001:** School characteristics in comprehensive schools (*n* = 2268).

**School Characteristics**		**Total (%)**
School type	1–6	1249 (61.2)
	7–9	234 (11.5)
	1–9	557 (27.3)
Type of municipality	Urban	1204 (53.1)
	Suburban	529 (23.3)
	Rural	535 (23.6)
Student count	Median	174
	Range	3–1374
	Q_1_–Q_3_	66–351

**Table 2 ijerph-19-12698-t002:** School violence in comprehensive in schools (*n* = 2268).

		Total (%)
Pupil threatened school personnel with violence or was violent towards them	No	773 (34.1)
	Yes	1223 (53.9)
	Unknown	272 (12.0)
Violence between pupils	No	388 (17.1)
	Yes	1613 (71.1)
	Unknown	267 (11.8)
Personnel’s inappropriate behavior towards a pupil (e.g., violence, harassment, bullying)	No	1588 (70.1)
	Yes	311 (13.7)
	Unknown	369 (16.3)

**Table 3 ijerph-19-12698-t003:** Number of schools regarding clusters and school violence (*n* = 2268).

		Cluster					
		1 No Violence	2 Adolescent Violence	3 Not Known	4 Peer Violence	5 All Violence	Total (%)
Pupil threatened school personnel with violence or was violent towards them	No	337	0	4	432	0	773 (34.1)
	Yes	44	851	1	0	327	1223 (53.9)
	Unknown	2	0	252	13	5	272 (12.0)
Violence between pupils	No	383	0	1	0	4	388 (17.1)
	Yes	0	842	4	440	327	1613 (71.1)
	Unknown	0	9	252	5	1	267 (11.8)
Personnel’s inappropriate behavior towards a pupil (e.g., violence, harassment, bullying)	No	352	851	3	382	0	1588 (70.1)
	Yes	22	0	2	44	243	311 (13.7)
	Unknown	9	0	252	19	89	369 (16.3)
Schools per cluster		383	851	257	445	332	2268 (100.0)
Student count	Median	63	247.5	123	126	351	174
	Range	3–997	6–1134	9–972	15–738	21–1374	3–1374
	Q_1_–Q_3_	39–117	126–408	51–270	57–270	188–521	66–351

## Data Availability

Public reports and summaries of the data can be found at teaviisari.fi. Municipal-level data can be obtained from an application programming interface, see https://thl.fi/fi/tilastot-ja-data/aineistot-ja-palvelut/avoin-data#TEAviisari (accessed on 28 October 2021). For school-level data please contact the authors.

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
