# Peer review of "Health Promotion Actions and School Violence—A Cluster Analysis from Finnish Comprehensive Schools"

_ijerph, 2022, doi:10.3390/ijerph191912698_

Round 1

Reviewer 1 Report

This is a very interesting study using data collected by the large-scale nationally representative BSHPCB aiming mainly to identify profiles of 267 Finnish schools based on school violence. According to the results of the cluster analysis, 5 distinct profiles of school violence have emerged: (1) No violence, (2) Adolescent violence, (3) Not known, (4) Peer violence, and (5) All violence. According to my opinion this study should be published to your J because it is important for your scientific audience. Authors are advised to take into consideration some minor comments.

Introduction

Reviewing school safety and violence, your references are too old and I wonder whether there are not more recent relevant papers that should be included.

Talking about the influence of some school characteristics on the violence, one would expect to see the role of school climate, the reactions of the school personnel as bystanders towards violent acts etc.

With respect to the "Aim", authors are invited to better describe some important terms, e.g., “profiles”, “promotion actions” “school characteristics”.

Measures

Although the authors list 8 subjects that were measured, it is however not clear how they were measured, e.g. school violence? under the subtitle "data analysis"-table 2, some indications are given which should be described to the corresponding measurement section.

The authors declare that they have used the data collected by BSHPCB. However, I believe that they should insert a relevant “Procedure” section and provide all the relevant information.

Discussion

Authors are advised to insert both, the strong and the weak points of their study.

Author Response

Thank you very much for your valuable comments and suggestions. We have revised the paper and hope that the manuscript is now clearer and understandable.

Best greetings,

Authors

Open Review

English language and style

( ) Extensive editing of English language and style required
( ) Moderate English changes required
( ) English language and style are fine/minor spell check required
(x) I don't feel qualified to judge about the English language and style

Yes

Can be improved

Must be improved

Not applicable

Does the introduction provide sufficient background and include all relevant references?

( )

(x)

( )

( )

Are all the cited references relevant to the research?

( )

(x)

( )

( )

Is the research design appropriate?

( )

(x)

( )

( )

Are the methods adequately described?

(x)

( )

( )

( )

Are the results clearly presented?

(x)

( )

( )

( )

Are the conclusions supported by the results?

(x)

( )

( )

( )

Comments and Suggestions for Authors

This is a very interesting study using data collected by the large-scale nationally representative BSHPCB aiming mainly to identify profiles of 267 Finnish schools based on school violence. According to the results of the cluster analysis, 5 distinct profiles of school violence have emerged: (1) No violence, (2) Adolescent violence, (3) Not known, (4) Peer violence, and (5) All violence. According to my opinion this study should be published to your J because it is important for your scientific audience. Authors are advised to take into consideration some minor comments.

Thank you for the encouraging comments.

Introduction

Reviewing school safety and violence, your references are too old and I wonder whether there are not more recent relevant papers that should be included.

We have revised the introduction and added more recent papers.

Talking about the influence of some school characteristics on the violence, one would expect to see the role of school climate, the reactions of the school personnel as bystanders towards violent acts etc.

Thank you for the valuable suggestion. We have added research on those issues.

With respect to the "Aim", authors are invited to better describe some important terms, e.g., “profiles”, “promotion actions” “school characteristics”.

We have described these more detailed.

Measures

Although the authors list 8 subjects that were measured, it is however not clear how they were measured, e.g. school violence? under the subtitle "data analysis"-table 2, some indications are given which should be described to the corresponding measurement section.

We have revised the text and tried to describe these clearer.

The authors declare that they have used the data collected by BSHPCB. However, I believe that they should insert a relevant “Procedure” section and provide all the relevant information.

We have revised this section and added “Procedure” section.

Discussion

Authors are advised to insert both, the strong and the weak points of their study.

We have added limitations and strengths.

Reviewer 2 Report

 The present review “Health promotion actions and school violence –a cluster analysis from Finnish comprehensive schools” proposes an interesting topic where there is a wide field to investigate.

I would have expected the review to mention alternative ways to prevent and solve the problem. What are schools doing to improve safety?

Preventing school violence requires addressing the factors that put people at risk or protect them from violence. The authors should explain the following item for example:

The role of parents’ involvement, train teachers and school staff on positive discipline, classroom management, and peaceful conflict resolution, develop and implement life skills and social and emotional learning programs to build the resilience and protective capacity of children and youth, and set up confidential and safe reporting mechanisms in schools.

Table 4 is not clear and transparent.

Conclusions need to be focused more.

Author Response

Thank you very much for your valuable comments and suggestions. We have revised the paper and hope that the manuscript is now clearer and understandable.

Best greetings,

Authors

Open Review

(x) I would not like to sign my review report

( ) I would like to sign my review report

English language and style

( ) Extensive editing of English language and style required

( ) Moderate English changes required

(x) English language and style are fine/minor spell check required

( ) I don't feel qualified to judge about the English language and style

This manuscript has been proofread and copy edited by the native professional.

                             Yes                      Can be improved                        Must be improved                     Not applicable

Does the introduction provide sufficient background and include all relevant references?

                             (x)                        ( )                         ( )                         ( )

Are all the cited references relevant to the research?

                             ( )                         (x)                        ( )                         ( )

Is the research design appropriate?

                             (x)                        ( )                         ( )                         ( )

Are the methods adequately described?

                             ( )                         (x)                        ( )                         ( )

Are the results clearly presented?

                             ( )                         ( )                         (x)                        ( )

Are the conclusions supported by the results?

                             (x)                        ( )                         ( )                         ( )

Comments and Suggestions for Authors

The present review “Health promotion actions and school violence –a cluster analysis from Finnish comprehensive schools” proposes an interesting topic where there is a wide field to investigate.

Thank you for the encouraging comment.

I would have expected the review to mention alternative ways to prevent and solve the problem. What are schools doing to improve safety?

Preventing school violence requires addressing the factors that put people at risk or protect them from violence. The authors should explain the following item for example:

The role of parents’ involvement, train teachers and school staff on positive discipline, classroom management, and peaceful conflict resolution, develop and implement life skills and social and emotional learning programs to build the resilience and protective capacity of children and youth, and set up confidential and safe reporting mechanisms in schools.

Thank you for this valuable comment. We have added WHO reference.

Table 4 is not clear and transparent.

We have revised the text and the table 4, and added more explanation for the Table 4.

Conclusions need to be focused more.

We have revised this section.